# The epidemiology of superficial Streptococcal A (impetigo and pharyngitis) infections in Australia: A systematic review

Sophie Wiegele[1,2]*, Elizabeth McKinnon[2], Bede van Schaijik[3], Stephanie Enkel[2,3], Katharine Noonan[2], Asha C. Bowen[1,2,3], Rosemary Wyber[2,3,4]

1 Perth Children's Hospital, Nedlands, Western Australia, Australia, 2 Telethon Kids Institute, Nedlands, Western Australia, Australia, 3 University of Western Australia, Perth, Western Australia, Australia, 4 National Centre for Epidemiology and Population Health, The Australian National University, Canberra, Australian Capital Territory, Australia

* Sophie.Wiegele@telethonkids.org.au

## Abstract

### Background

Streptoccocal A (Strep A, GAS) infections in Australia are responsible for significant morbidity and mortality through both invasive (iGAS) and post-streptococcal (postGAS) diseases as well as preceding superficial (sGAS) skin and throat infection. The burden of iGAS and postGAS are addressed in some jurisdictions by mandatory notification systems; in contrast, the burden of preceding sGAS has no reporting structure, and is less well defined. This review provides valuable, contemporaneous evidence on the epidemiology of sGAS presentations in Australia, informing preventative health projects such as a Streptococcal A vaccine and standardisation of primary care notification.

### Methods and findings

MEDLINE, Scopus, EMBASE, Web of Science, Global Health, Cochrane, CINAHL databases and the grey literature were searched for studies from an Australian setting relating to the epidemiology of sGAS infections between 1970 and 2020 inclusive. Extracted data were pooled for relevant population and subgroup analysis. From 5157 titles in the databases combined with 186 grey literature reports and following removal of duplicates, 4889 articles underwent preliminary title screening. The abstract of 519 articles were reviewed with 162 articles identified for full text review, and 38 articles identified for inclusion. The majority of data was collected for impetigo in Aboriginal and Torres Strait Islander populations, remote communities, and in the Northern Territory, Australia. A paucity of data was noted for Aboriginal and Torres Strait Islander people living in urban centres or with pharyngitis. Prevalence estimates have not significantly changed over time. Community estimates of impetigo point prevalence ranged from 5.5–66.1%, with a pooled prevalence of 27.9% [95% CI: 20.0–36.5%]. All studies excepting one included >80% Aboriginal and Torres Strait Islander people and all excepting two were in remote or very remote settings. Observed prevalence of impetigo as diagnosed in healthcare encounters was lower, with a pooled estimate of

This is a Registered Report and may have an associated publication; please check the article page on the journal site for any related articles.

**Data Availability Statement:** All relevant data are within the paper and its Supporting Information files.

**Funding:** Incidental costs for this paper were supported by the NHMRC funded END RHD CRE (1080401). RW was supported an NHMRC Postgraduate Scholarship (1151165) and AB by an NHMRC Investigator Award (GNT1175509).

**Competing interests:** The authors have declared that no competing interests exist.

10.6% [95% CI: 3.1–21.8%], and a range of 0.1–50.0%. Community prevalence estimates for pharyngitis ranged from 0.2–39.4%, with a pooled estimate of 12.5% [95% CI: 3.5–25.9%], higher than the prevalence of pharyngitis in healthcare encounters; ranging from 1.0–5.0%, and a pooled estimate of 2.0% [95% CI: 1.3–2.8%]. The review was limited by heterogeneity in study design and lack of comparator studies for some populations.

## Conclusions

Superficial Streptococcal A infections contribute to an inequitable burden of disease in Australia and persists despite public health interventions. The burden in community studies is generally higher than in health-services settings, suggesting under-recognition, possible normalisation and missed opportunities for treatment to prevent postGAS. The available, reported epidemiology is heterogeneous. Standardised nation-wide notification for sGAS disease surveillance must be considered in combination with the development of a Communicable Diseases Network of Australia (CDNA) Series of National Guideline (SoNG), to accurately define and address disease burden across populations in Australia.

## Trial registration

This review is registered with PROSPERO. Registration number: CRD42019140440.

## Introduction

*Streptococcus pyogenes* (GAS, Strep A) is an obligate human pathogen causing an array of superficial (sGAS), invasive (iGAS) and post-infectious complications (postGAS) (Fig 1) [1–3]. Australia has a high burden of GAS infection across all disease endpoints, all of which disproportionately affect Aboriginal and Torres Strait Islander people [4–9]. A robust understanding of GAS prevalence and incidence is essential to address this disparity and inform preventative strategies including primary care notification, enhanced environmental health strategies and the development of a Strep A vaccine. At present, the most urgent epidemiologic need is an improved understanding around the burden of sGAS infections of the skin (impetigo) and throat (pharyngitis) which are the two most likely concurrent drivers of acute rheumatic fever (ARF). GAS is known to be the principal pathogen in tropical impetigo with isolates detected from 80–90% of lesions [10,11], and a significant contributor to pharyngitis with GAS culture positive in 20% of presentations [12]. These represent a large reservoir of infection and transmission leading to iGAS and postGAS outcomes. However, people with these sGAS infections may not seek medical care and, if they do, are usually treated in primary care where epidemiologic data is not routinely aggregated. Additionally, most of these infections are managed without microbial confirmation of the pathogen which precludes use of laboratory data for understanding disease burden. Therefore, sGAS burden estimates are largely based on academic research, or extrapolated from studies focused on the clinical phenotypes of *probable* sGAS disease. A review focused on the of clinical presentations likely to represent sGAS disease (impetigo and pharyngitis) is more likely to capture the burden of disease, likely to be under-represented in studies requiring microbiological confirmation given current best practice in Australia. Past reviews have generally focused on *either* impetigo or pharyngitis alone in discrete geographic areas. A number of reviews have synthesised these results for impetigo [6,13], but pharyngitis epidemiology has not been a focus of systematic review

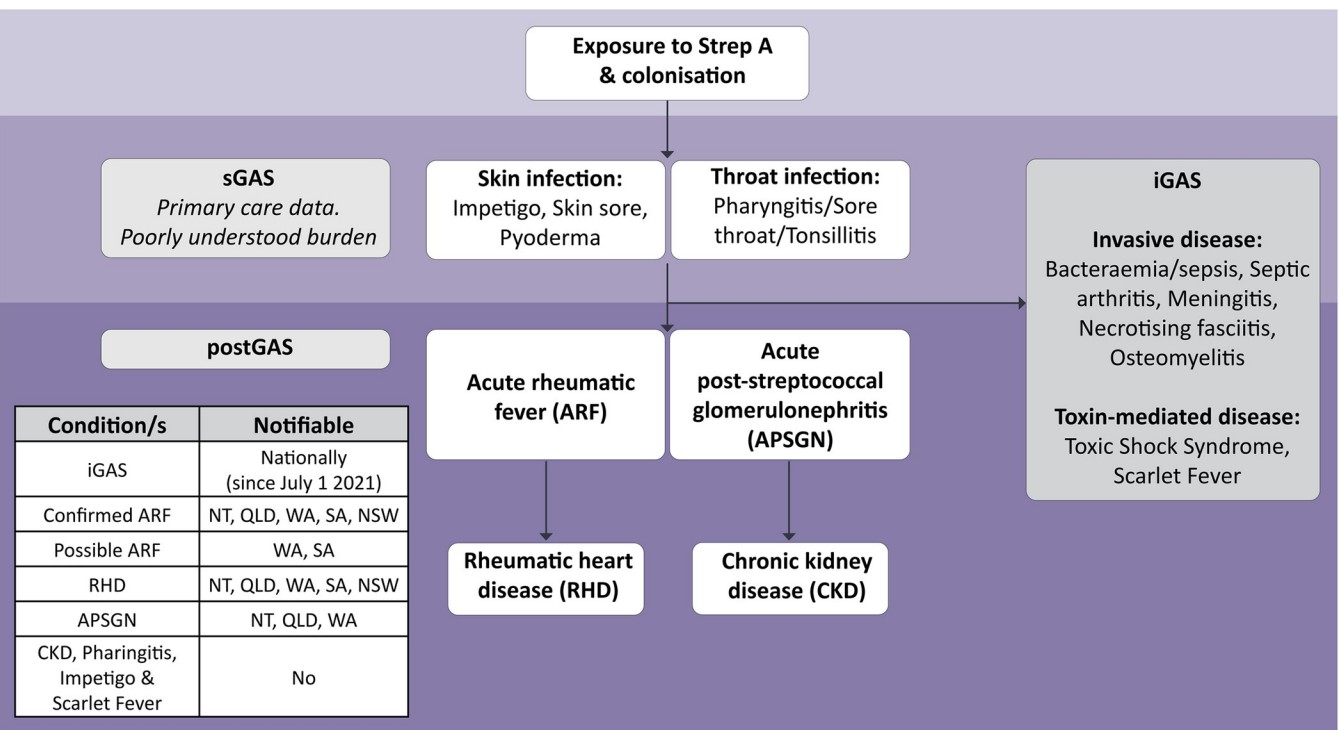

**Fig 1. Spectrum of Streptococcal A related disease and notification status in Australia.** The nomenclature for sGAS also remains diverse and the various terms used to describe this have been included in the figure for both skin and throat infection.

globally, with no study presenting pharyngitis data specific to an Australian context [14]. No review describes the burden of disease in both impetigo and pharyngitis presentations, although both are equally linked to iGAS and postGAS outcomes. This impedes a clear national understanding of sGAS and the disparities between different populations. In contrast, iGAS was made nationally notifiable in July 2021 [15] and many postGAS endpoints are notifiable at a jurisdictional level (Fig 1), providing some longitudinal visibility of the severe GAS disease burden [4]. A systematic approach to aggregating estimates of skin and throat sGAS burden is needed to provide a foundation for unified burden of disease estimates, identify data gaps and inform future planning.

In this systematic review, we aimed to provide a comprehensive overview of sGAS skin and throat infections in different populations in Australia over the past five decades. In particular, we aimed to provide a composite picture of the burden of disease in different settings accounting for differences in case ascertainment.

## Methods

This systematic review is reported according to the updated Preferred Reporting items for Systematic Reviews and Meta-Analyses (PRISMA) statement and the Meta-analysis of Observational Studies in Epidemiology (MOOSE) guidelines [16–18]. The search strategy and review protocol have been published elsewhere, and no amendments have been made [19]. This review included published and unpublished articles containing epidemiological data (incidence and prevalence) on the clinical presentations of a probable sGAS infection (impetigo/ skin sore/ pyoderma and/or pharyngitis/ tonsillitis) in English from the years 1970–2020 inclusive from Australian settings. Articles were excluded if there were no data on the clinical

presentation of the patient, epidemiology presented was from a baseline population with potential confounding factors such as other infectious diseases, or the terms used to describe clinical presentation were vague or too general.

There were difficulties in accurately classifying infections compounded by the non-standard nomenclature for skin and throat infections, including impetigo, skin sore, purulent skin sore, pyoderma and sore throat, tonsillitis, pharyngitis respectively. Included terms were defined conservatively. For example, 'upper respiratory tract infection' and 'skin infection' were considered too broad and unlikely to reflect a *probable* sGAS infection in general, therefore were excluded [20–24]. The term 'infected scabies' is likely to represent co-infection of impetigo and scabies, therefore such cases contributed to prevalence estimates if provided. For the purposes of this report we have standardised nomenclature to 'impetigo' for a *probable* sGAS skin infection and 'pharyngitis' for a *probable* sGAS throat infection.

This review executed search strategies through MEDLINE, Scopus, EMBASE, Web of Science, Global Health, Cochrane and CINAHL as well as a comprehensive grey literature search [19]. The final database search was executed in March 2021. Search strategies were developed and executed by the research team with assistance from University of Western Australia librarians. Title screen was completed by one independent investigator in consultation with senior authors during the process of data familiarisation, with two independent investigators completing abstract and full text review for each included paper (BvS, EM and/or SW).

Data was extracted from each study into an online template form by two independent authors (BvS, EM, SE, and/or SW). Extracted data was then compared and discrepancies discussed between the authors. All data points with definitions are described in the protocol paper [19]. Data collection templates and extracted data are available through contact with the corresponding author (SW).

Overall study quality was assessed for each paper using the Joanna Briggs Institute critical appraisal checklist [25]. This included identification and assessment of biases, study power and methods. This review was completed with careful consideration of reduction in meta-biases. Publication bias was reduced by a rigorous search of the grey literature and unpublished articles. The search spanned a 50-year publication period (1970–2020) and combined with the persisting nature of GAS disease this worked to reduce the risk of time-lag bias and allowed for changes in epidemiological data over time. Careful scrutiny for duplication of datasets, tailored search strategies across several major databases and grey literature sources, as well as inclusion of all formats further decreased bias. Inclusion of articles was based on meeting predetermined criteria irrespective of primary outcome.

Prevalence and/or incidence estimates of impetigo and pharyngitis were extracted or derived from each eligible publication. Data characterizing population subgroups (remoteness, climate type, Aboriginal and Torres Strait Islander status, age strata) and study setting were recorded. For presentation of results, studies have been categorized into community or healthcare settings, according to the denominator of the prevalence estimate. Community settings were those where people in an area, town or school were actively screened for presence of disease. Population prevalence estimates in these settings were obtained from the number of cases as a proportion of participants screened. Point prevalence estimates were used if possible; where disease burden was tracked over time by multiple visits to a community, the first prevalence estimate has been used, if available. Healthcare settings comprise those studies where data was drawn from utilisation of a health service. These include attendance at a community health clinic or general practice, presentation at an emergency department, and inpatient admission. For these settings prevalence estimates reflect the proportion of health service encounters where impetigo or pharyngitis was diagnosed. In those instances where multiple visits to the health service were made for a single episode of infection, studies typically noted

this was recorded as a single encounter. Incidence rates related to impetigo and pharyngitis are reported for those studies where they have been provided. These are expressed in events (diagnoses or admissions) per 100 person-years of follow-up.

Analyses have considered impetigo and pharyngitis data separately for both community settings and healthcare encounters. Pooled prevalence estimates have been obtained from random-effects meta-analyses, with effect sizes based on the double-arcsine transform of the raw proportions and analyses weighted for inverse variance using the restricted maximum-likelihood method. Heterogeneity of estimates across studies is quantified with the $I^2$ statistic and assessed with the Cochran Q statistic. As pre-planned, subgroup analysis and meta-regression has been used to explore variation in prevalence data according to: geography (urban, remote), Aboriginal and Torres Strait Islander status, climate type, age of population and year of study. Asymmetry of standard errors against effect sizes are considered by a visual examination of funnel plots, and assessed by Begg's rank correlation test. Data were analysed in R version 4.02 (R Project for Statistical Computing,Vienna, Austria http://www.R-project.org) within the RStudio integrated development environment (RStudio Team, Boston MA).

## Results

The initial search resulted in 5157 database results that were combined with 186 results from the grey literature review and hand search. After duplicates were removed, 4889 articles underwent preliminary title screening with 4370 articles excluded. The abstract of 519 articles were reviewed with 162 articles identified for full text review. One article could not be sourced. Thirty-eight articles presenting 40 studies were included after full text review by two independent reviewers (Fig 2) [1,26–62]. A description of included studies can be visualised in the supporting material (S1 Table).

Of the 38 included articles, 28 addressed impetigo (74%), with the majority of these reporting on Aboriginal and Torres Strait Islander populations (25/28, 89%) and with data more often collected in a community setting (20/28, 71%) (Fig 3). Seventeen articles (45%) included the epidemiology of pharyngitis, with more focus on non-Indigenous populations (9/17, 57%). Only seven articles addressed both impetigo and pharyngitis, all of which were in Aboriginal and Torres Strait Islander populations.

Fig 3B provides a visual representation of the geographical distribution of included articles. The largest representation was for the Northern Territory, particularly Arnhem Land (n = 21/ 38, 55%). The least data, based on geography, was available for the Australian Capital Territory and Tasmania, with only the Australia-wide articles including these geographical regions (n = 4).Despite search strategy implementation over a fifty-year period (1970–2020), all included studies but one (Pollard et al) published data collected within the last 30 years [54]. The years of data collection vary from 1971–2016 with the majority of studies collecting data from 2000 to 2020 (29/40, 73%) (Fig 3C). There was no statistically significant trend in the burden of impetigo or pharyngitis in either setting over time.

### Impetigo

Prevalence data could be extracted or derived from the 28 articles that addressed impetigo. These covered 29 separate studies, 20 within the community setting (Fig 4) and 9 in a healthcare setting (Fig 5), with one article reporting on multiple data sets [62].

Estimates of impetigo point prevalence within the community setting ranged from 5.5–66.1%, with a pooled prevalence of 27.9% [95% CI: 20.0–36.5%] (Fig 4). Observed prevalence of impetigo as diagnosed in a healthcare setting was lower, with a pooled prevalence of 10.6% [95% CI: 3.1–21.8%] obtained from a prevalence range of 0.1–50.0% over the nine studies

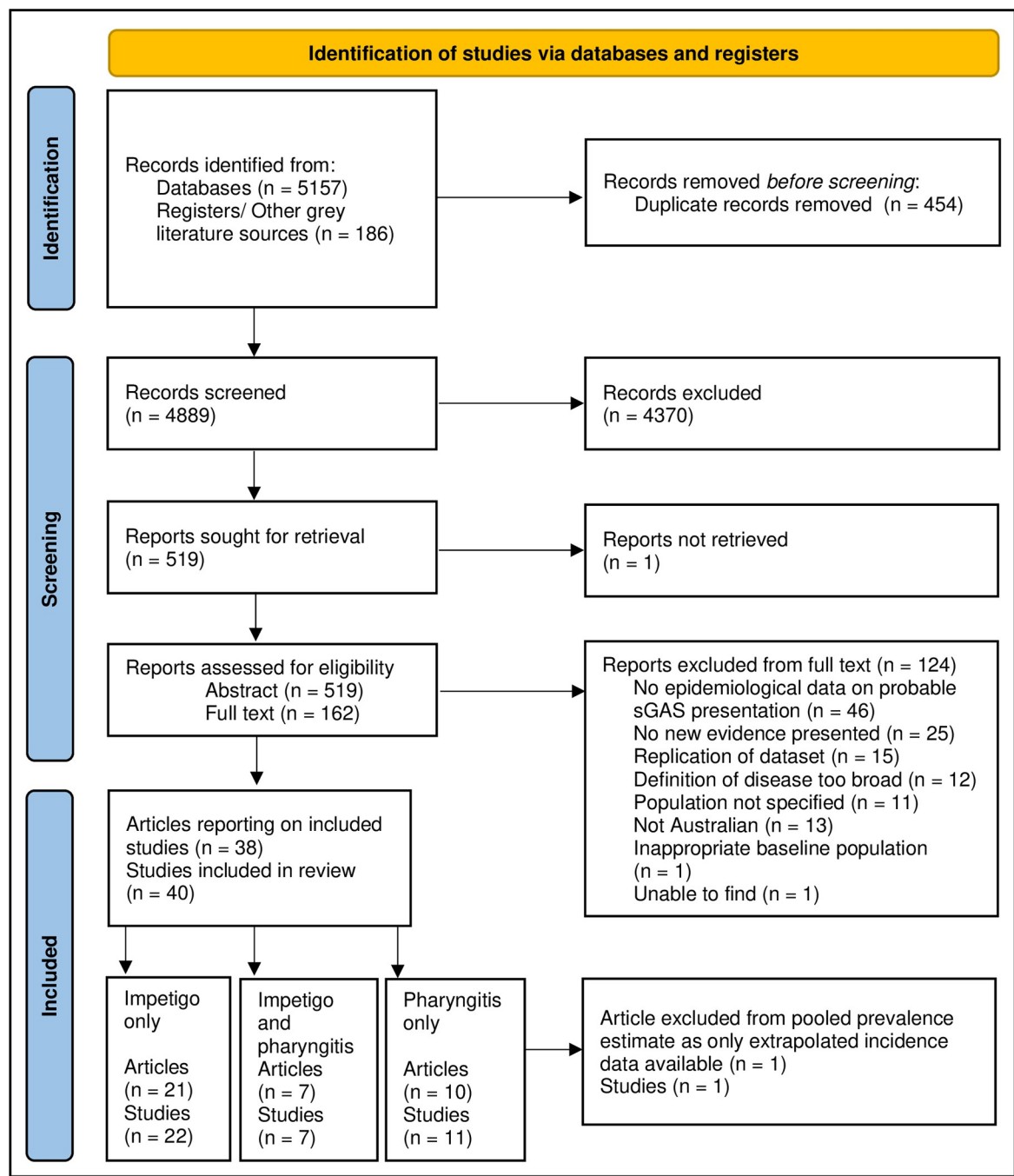

**Fig 2. PRISMA flow chart.** Reported according to the PRISMA Protocol for the reporting of systematic reviews [16,18].

included (Fig 5). Fewer of these studies represented a predominantly Aboriginal and Torres Strait Islander population (n = 5).

Considerable between-study variability was observed in the prevalence estimates, with $I^2$ greater than 99% (p<0.001) for each of the two settings. Subgroup analysis demonstrated an increase in impetigo prevalence associated among studies conducted wholly in a tropical region when considering diagnoses in a healthcare setting (p = 0.004, Table 1), but no

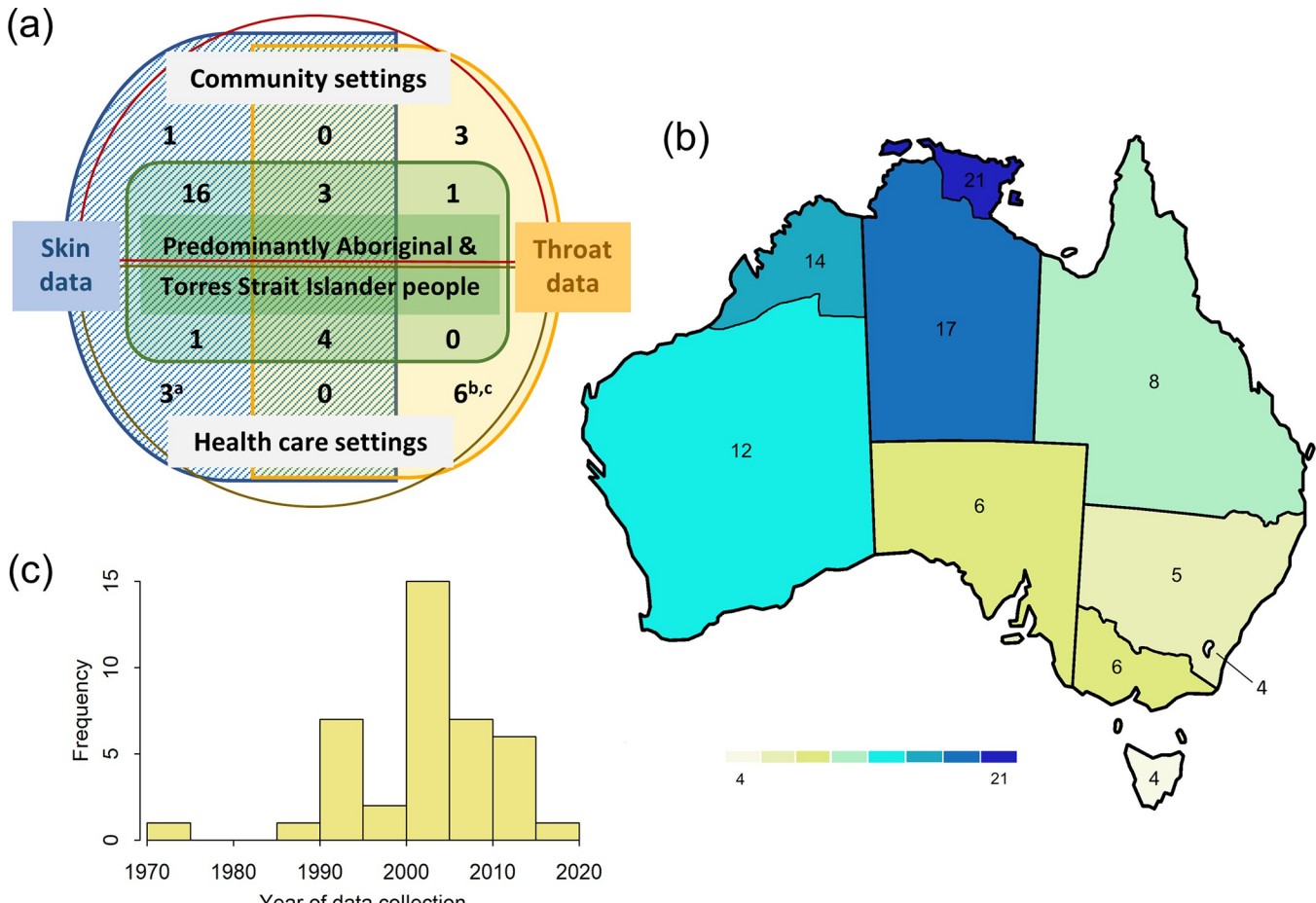

**Fig 3. Demographics of included articles.** (a) Population representation in included articles. *Tan: Reporting only pharyngitis. Tan/blue overlay: Reporting on both impetigo and pharyngitis. Blue: Reporting on only impetigo. Green: Reporting on ≥80% First Nations population.* [a]*Two studies were reported in Yeoh et al (2017) [62], with data collected prospectively and retrospectively;* [b]*Two studies were reported in Charles et al (2004) [32], with data collected a decade apart;* [c]*Del Mar (1995) [36], included only extrapolated incidence data and was omitted from analysis of prevalence.* (b) Geographical coverage of included articles. *Number of articles covering specified territories within Australia; Western Australia excluding the Kimberley region, Western Australia including the Kimberley region, Northern Territory excluding Arnhem Land, Northern Territory including Arnhem Land, South Australia, Victoria, Tasmania, New South Wales, Australian Capital Territory and Queensland.* (c) Year/mid-point year of data collection of included studies (n = 40).

differences across climate type in general prevalence in the community. Meta-regression did not find any other statistically significant moderator effects to explain the observed heterogeneity independent of climate (proportion of sample living rurally: p = 0.01 when considered in univariate analysis, but p = 0.8 when considered jointly with tropical climate). However, Heath et al., a study of school-children in a large regional tropical town, documented increased prevalence of impetigo observed in Aboriginal and Torres Strait Islander participants compared with non-Indigenous participants (22.2% [95% CI: 13.0–32.0%] vs 5.9% [95% CI: 0.0–11.0%]) [38].

Only a single paper reported population-level incidence using linked data [1]. This described paediatric hospital admissions where impetigo was diagnosed as a primary or secondary condition. The authors note markedly higher hospital admission rates for Aboriginal and Torres Strait Islander children with a diagnosis of impetigo across all age strata (incident rate ratio of Aboriginal and Torres Strait Islander to non-Indigenous admissions (95%CI), per 1000 child-years: <1 year 24.9 (22.0±28.2); 1–4 years 22.0 (19.8±24.5); 5–9 years 16.3 (13.4

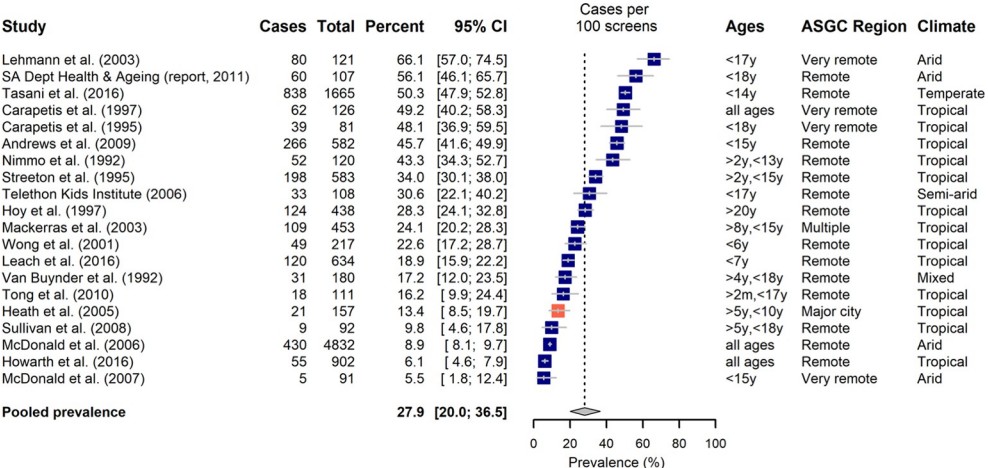

**Fig 4. The prevalence of impetigo diagnosed in a community setting, categorized by predominance of Aboriginal and Torres Strait Islander people in the study.** Aboriginal and Torres Strait Islander representation in study: Blue >80%, Tomato 20–80%. Climate: Reported according to Koppen Classification System [63,64].

±19.9); 10–15 years 14.8 (10.5±20.7)). Of note, McMeninan et al. reported a significant cumulative incidence of 82% of children with at least one episode of impetigo in the first year of life.

Half of the articles included in our analyses presented data on both impetigo and scabies infection. Rates of co-infection were noted in five of these [26,50,51,58,61], with scabies present in 3% - 51% of impetigo cases. Nine articles reported scabies prevalence only, separately to that of impetigo [33,39,41,42,44–46,48,62], and fourteen articles did not report on scabies infection.

## Pharyngitis

Pharyngitis prevalence estimates could be extracted or derived from 17 studies. In contrast to impetigo articles, more studies examining prevalence of pharyngitis were carried out in healthcare settings (n = 10, with two datasets presented by Charles et al [32]) than were conducted by community sampling (n = 7). One paper was excluded from pooled prevalence analysis as only incidence data, extrapolated from general practice attendance, was available [36].

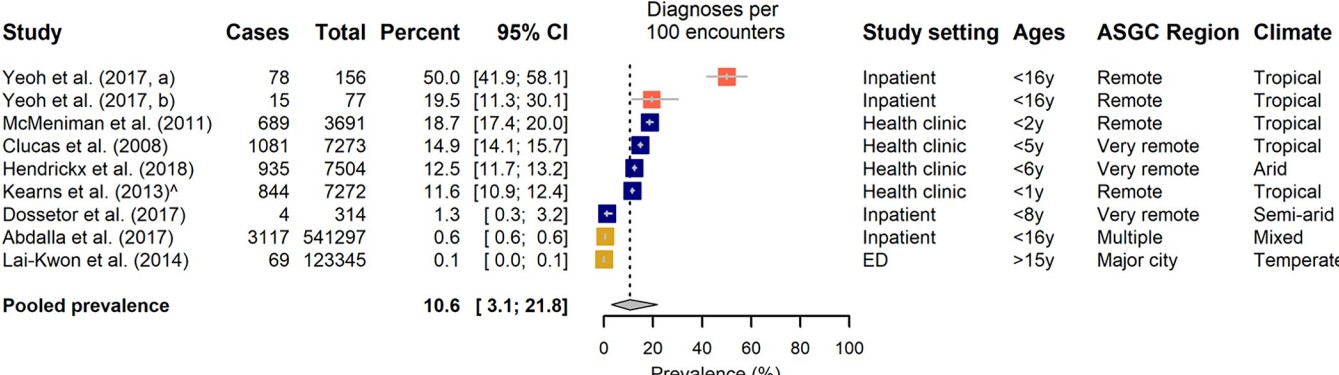

**Fig 5. The prevalence of impetigo diagnosed in a healthcare setting, categorized by predominance of Aboriginal and Torres Strait Islander people in the study.** Aboriginal and Torres Strait Islander representation in study: Blue >80%, Tomato 20–80%, Mustard <20%. Climate: Reported according to Koppen Classification System [63,64].

**Table 1. Prevalence estimates for studies of impetigo across population subtypes.**

| | | Prevalence percentage | | | |
|---|---|---|---|---|---|
| Study group | N | Range | Pooled estimate [95% CI] | I$^2$ | P-value |
| **Community setting** | | | | | |
| Overall | 20 | 5.5–66.1 | 27.9 [20.0, 36.5] | 99.0% | |
| Climate | | | | | 0.99 |
| Tropical | 14 | 6.1–50.3 | 27.9 [18.4, 38.5] | 98.6% | |
| Non-tropical | 6 | 5.5–66.1 | 27.9 [13.9, 44.6] | 98.6% | |
| Aboriginal and Torres Strait Islander Status | | | | | 0.38 |
| Non-Indigenous | 1 | 13.4–13.4 | 13.4 [0.0, 50.6] | - | |
| Aboriginal and Torres    Strait Islander People | 19 | 5.5–66.1 | 28.8 [20.6, 37.7] | 99.1% | |
| Region | | | | | 0.43 |
| Mixed/Urban | 2 | 13.4–24.1 | 18.5 [2.2, 45.3] | 88.4% | |
| Remote/Very remote | 18 | 5.5–66.1 | 29.0 [20.6, 38.3] | 99.1% | |
| Age | | | | | 0.34 |
| Adults/Mixed ages | 4 | 6.1–49.2 | 20.4 [6.8, 38.9] | 98.7% | |
| Children (>80%) | 16 | 5.5–66.1 | 29.9 [20.9, 39.8] | 97.4% | |
| **Health setting** | | | | | |
| Overall | 9 | 0.1–50.0 | 10.6 [3.1, 21.8] | 99.9% | |
| Climate | | | | | 0.002 |
| Tropical | 5 | 11.6–50.0 | 21.6 [11.1, 34.4] | 98.0% | |
| Non-tropical | 4 | 0.1–12.5 | 2.1 [0.0, 8.9] | 99.9% | |
| Aboriginal and Torres Strait Islander status | | | | | 0.99 |
| Non-Indigenous | 4 | 0.1–50.0 | 10.5 [0.5, 29.9] | 99.8% | |
| Aboriginal and Torres    Strait Islander people | 5 | 1.3–14.9 | 10.8 [1.2, 27.7] | 98.0% | |
| Region | | | | | 0.01 |
| Mixed/Urban | 2 | 0.1–0.6 | 0.3 [0.0, 8.6] | 99.9% | |
| Remote/Very remote | 7 | 1.3–50.0 | 16.1 [7.5, 27.2] | 98.0% | |
| Age | | | | | 0.13 |
| Adults/Mixed ages | 1 | 0.1–0.1 | 0.06 [0.0, 18.8] | - | |
| Children (>80%) | 8 | 0.6–50.0 | 13.3 [4.7, 24.9] | 99.9% | |

N = number of studies.

In the community setting prevalence estimates for pharyngitis ranged from 0.8–39.4%, with a pooled estimate of 12.5% [95% CI: 3.5–25.9%] (Fig 6). Prevalence of diagnoses amongst health service encounters was in general considerably lower, ranging from 1.0–5.0%, with a

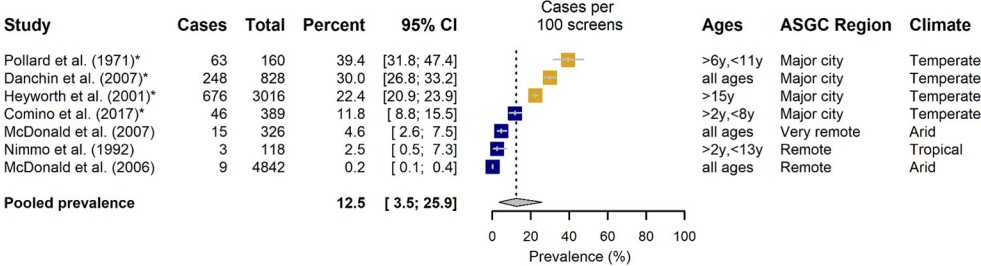

**Fig 6. The prevalence of pharyngitis diagnosed in a community setting, categorized by predominance of Aboriginal and Torres Strait Islander people in the study.** Aboriginal and Torres Strait Islander representation in study: Blue >80%, Mustard <20%. Climate: Reported according to Koppen Classification System [63,64].

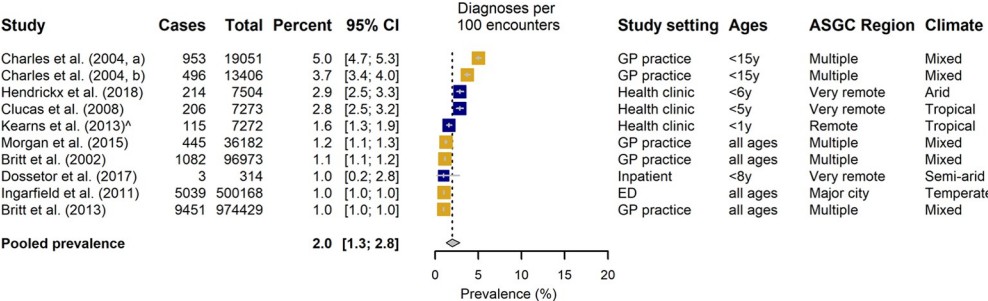

**Fig 7. The prevalence of pharyngitis diagnosed in a healthcare setting, categorized by predominance of Aboriginal and Torres Strait Islander people in the study.** Aboriginal and Torres Strait Islander representation in study: Blue >80%, Mustard <20%. Climate: Reported according to Koppen Classification System[63,64].

pooled estimate of 2.0% [95% CI: 1.3–2.8%] (Fig 7). Overall, there were fewer studies that focussed predominantly on Aboriginal and Torres Strait Islander populations, particularly in the healthcare settings.

Considerable heterogeneity of study estimates was again noted for both settings ($I^2$ >99%) (Table 2). Prevalence of pharyngitis was found to be significantly lower in community-based studies that had predominantly Aboriginal and Torres Strait Islander representation compared with studies undertaken within a general Australian population (3.6% [95% CI: 0.5–9.2%] vs 30.0% [18.7–42.7%], p<0.0001) (Table 2). However, it should be noted that all studies in the latter group presented only period prevalence estimates of pharyngitis, and there was no indication of differential rates reported in a health care setting. Analysis that considered remoteness of study location reflected a confounding of this variable with Aboriginal and Torres Strait Islander representation. One state-wide study that surveyed South Australia households reported comparable pharyngitis prevalence across rural and metropolitan locations (20.9% (95%CI: 17.8–24.0%) and 22.9% (21.2–24.6%) respectively) [40]. In healthcare settings, subgroup analysis of studies dichotomized by age range of participants found that a diagnosis of pharyngitis was more prevalent in studies which focussed on children (p = 0.0007, Table 2). This finding was confirmed in the meta-regression analysis that considered as a continuous covariate the proportion of study participants aged <18 years (p = 0.0009), moreover increased prevalence of diagnosed throat infection was associated with a greater proportion of school-aged children (p<0.0001).

Four studies provided estimates of incidence rates for pharyngitis [35,36,49,50] (Table 3). On average, incidence of proven GAS infection was approximately 25% of those reporting pharyngitis. The greater childhood incidence rates of sore throat observed in the prospective Melbourne cohort study [35] and sentinel GP data across the state of Queensland [36] were not observed in the Aboriginal and Torres Strait Islander community surveillance studies [49,50].

Funnel plot analysis did not find significant asymmetry in plots of standard error against effect size (S1 Fig), nor were there significant trends of effect size by year of publication for any of the settings considered.

## Discussion

This metanalysis demonstrates that the prevalence estimates for sGAS disease have been relatively static in studies conducted over the last 50 years (1970–2020) and that there is a persistently high reported burden of disease in remote settings and among Aboriginal and Torres Strait Islander people in Australia. Overall, the prevalence of impetigo across healthcare and

**Table 2. Prevalence estimates for studies of pharyngitis across population subtypes.**

| Study group | N | Prevalence percentage | | I² | P-value |
|---|---|---|---|---|---|
| | | Range | Pooled estimate [95% CI] | | |
| **Community setting** | | | | | |
| Overall | 7 | 0.2–39.4 | 12.5 [3.5, 25.9] | 99.7% | |
| Climate | | | | | 0.39 |
| Tropical | 1 | 2.5–2.5 | 2.5 [0.0, 35.7] | - | |
| Non-tropical | 6 | 0.2–39.4 | 14.6 [4.0, 30.3] | 99.8% | |
| Aboriginal and Torres Strait Islander status | | | | | <0.0001 |
| Non-Indigenous | 3 | 22.4–39.4 | 30.0 [18.7, 42.7] | 94.5% | |
| Aboriginal and Torres Strait Islander people | 4 | 0.2–11.8 | 3.6 [0.5, 9.2] | 98.2% | |
| Region | | | | | <0.0001 |
| Mixed/Urban | 4 | 11.8–39.4 | 25.0 [15.5, 35.8] | 96.0% | |
| Remote/Very remote | 3 | 0.2–4.6 | 1.8 [0.0, 7.5] | 95.4% | |
| Age | | | | | 0.73 |
| Adults/Mixed ages | 4 | 0.2–30.0 | 10.7 [0.7, 29.9] | 99.8% | |
| Children (>80%) | 3 | 2.5–39.4 | 15.1 [1.1, 40.1] | 97.4% | |
| **Health setting** | | | | | |
| Overall | 10 | 1.0–5.0 | 2.0 [1.3, 2.8] | 99.5% | |
| Climate | | | | | 0.83 |
| Tropical | 2 | 1.6–2.8 | 2.2 [0.6, 4.5] | 96.3% | |
| Non-tropical | 8 | 1.0–5.0 | 1.9 [1.1, 3.0] | 99.6% | |
| Aboriginal and Torres Strait Islander status | | | | | 0.85 |
| Non-Indigenous | 6 | 1.0–5.0 | 1.9 [1.0, 3.1] | 99.7% | |
| Aboriginal and Torres Strait Islander people | 4 | 1.0–2.9 | 2.1 [0.9, 3.7] | 92.4% | |
| Region | | | | | 0.85 |
| Mixed/Urban | 6 | 1.0–5.0 | 1.9 [1.0, 3.1] | 99.7% | |
| Remote/Very remote | 4 | 1.0–2.9 | 2.1 [0.9, 3.7] | 92.4% | |
| Age | | | | | 0.0007 |
| Adults/Mixed ages | 4 | 1.0–1.2 | 1.1 [0.6, 1.8] | 92.3% | |
| Children (>80%) | 6 | 1.0–5.0 | 2.8 [2.1, 3.7] | 98.0% | |

N = number of studies.

community settings is two to five times higher than pharyngitis, in keeping with current understanding of impetigo as the prominent preceding infection for iGAS and postGAS immunological sequelae in Australia [2]. The predominance of impetigo was particularly stark among Aboriginal and Torres Strait Islander community settings, with pooled point prevalence of 28.8% [95% CI: 20.6–37.7%]. This estimate is lower than previously published works, based on slightly different inclusion criteria which incorporated more grey literature [6,13].

Impetigo and pharyngitis prevalence in a community setting was highest for both (27.9% and 12.5%) when compared to healthcare settings (10.6% vs 2.0%). This provides some indication of the magnitude of likely under-detection of sGAS infection: at a population level there may be 2.5 times as many impetigo infections in the community as are diagnosed in clinics, and 7.3 times as many pharyngitis episodes. Although comparing different studies between community and health care settings is associated with considerable uncertainty these results are supported by a number of other observations about impetigo. Firstly, one study has explored recognition of impetigo burden in a healthcare setting and indicated an under-recognition gap of a similar magnitude (49.5% on active screening at the time of hospital admission

**Table 3. Estimated incidence rates of pharyngitis in included studies.**

| Study | Study description | Age strata (years) | Events [95% CI] per 100 years follow-up | |
| --- | --- | --- | --- | --- |
| | | | Episodes of sore throat | Culture positive GAS infection |
| Danchin, 2007 [35] | Prospective, family-based metropolitan cohort study | <5 | 38.1 [28.5–47.8] | 10.3 [4.9–19.0] |
| | | 5–12 | 32.9 [28.3–37.6] | 12.8 [9.5–16.8] |
| | | 13–18 | 40.0 [28.1–51.9] | 9.2 [3.4–20.1] |
| | | >20 | 13.5 [10.2–16.8] | 4.7 [2.9–7.3] |
| Del Mar, 1995 [36] | Sentinel GP practice data (state-wide, Qld) | 0–4 | 18.0 | |
| | | 5–15 | 16.8 | |
| | | 16–20 | 18.2 | |
| | | 20–70 | 8.0 | |
| McDonald, 2006 [50] | Surveillance of First Nations community (Top End, NT); 49 households, 1173 participants | all ages | 19 [12–28] | 4 [1.4–15.5] |
| McDonald, 2007 [49] | Surveillance of First Nations community (Central Australia, NT); 13 households, 145 participants | all ages | 480 [290–780] | 32 [8–77] |

vs 19.5% on retrospective review of medical records) confirming their hypothesis that skin infections are normalised by health care practitioners when the burden of skin infections are high [62]. Secondly, high community prevalence of impetigo is consistent over time and across different settings, there remains uncertainty about this for pharyngitis. Thirdly, qualitative work has identified a range of reasons that people with impetigo may not seek health services, including stigma, inadequate access to acceptable primary care services, logistics, costs, normalisation, fear and racism [65–67]. Similarly, people who do attend for care may not have their infections appropriately recorded or treated because impetigo is normalised or there is insufficient health provider awareness of their risks [62,65]. Collectively, these factors represent reasonable evidence of a prevalence gap between community and health care settings; this metanalysis provides some indication of the scale of that gap on an aggregate basis but further research is required to improve confidence in this estimate and explore potential variation in different settings. Understanding this gap is important because sGAS disease not identified in healthcare settings is a missed opportunity to deliver antibiotic primary prevention which can reduce the attack rate of ARF following Strep A infection by up to 90% [68]. Recognition and treatment of impetigo to prevent ARF is now incorporated in the Rheumatic Heart Disease Australia (RHDA) guidelines as a high priority [69]. Understanding and addressing the under-recognition of sGAS is essential to reducing the inequitable burden of ARF and RHD among Aboriginal and Torres Strait Islander people [70].

Prevalence data from the Northern Territory, Aboriginal and Torres Strait Islander people and remote settings were over-represented in included studies. There was insufficient statistical power to compare the pooled prevalence of Aboriginal and Torres Strait Islander people in impetigo analysis in either healthcare or community settings. There were relatively few studies of sGAS among Aboriginal and Torres Strait Islander people outside the Northern Territory despite more than half of Aboriginal and Torres Strait Islander people living in Queensland or New South Wales [71]. Little is known about this sGAS burden. There were no significant differences in prevalence between mixed/urban settings and remote settings other than for pharyngitis in community settings, which revealed a high burden in mixed/urban location (pooled prevalence 25.0% (95% CI 15.5–35.8) relative to remote settings (pooled prevalence 1.8%, 95% CI 0.00–7.5) [p<0.0001]. The lack of significant data in other groups may reflect the low number of comparator studies in urban populations. Similarly, climate type had a statistically

significant influence on only one population: impetigo in a healthcare setting. There was no significant change in disease burden between tropical and non-tropical regions in all other settings. Strategies to address knowledge gaps for specific geographies and populations are needed to equitably address sGAS burden.

Visual inspection of the forest plots for prevalence of impetigo in a healthcare setting has a clear statistical outlier, Yeoh et al (2017) which may be explained by the data collection methods; specifically prospective data collection of the skin examination on all children admitted to hospital to address their hypothesis that skin infections were being normalised and under-documented. Clearly this was confirmed when compared to the retrospective chart review of skin infection documentation in the previous year for a similar cohort, and may well represent the gap in documentation or under-ascertainment bias for all other included studies [62]. Similarly, Lehmann et al. also reported a significantly higher burden of impetigo; potentially in association with a concurrent scabies outbreak in the community at the time of data collection [47]. It is clear that scabies infection is a major driver of impetigo, and where available this review has included impetigo prevalence inclusive of scabies co-infection data. However, this was significantly limited by either non-report of scabies status (n = 14) or use of generalised terms; 'total scabies' and 'total impetigo' (n = 9).

Few studies reported on microbiological data from the patient cohort for either impetigo [50,53,56,59,60,62] (n = 6) or pharyngitis [35,43,50] (n = 3) disease resulting in insufficient power for statistical analysis based on GAS+ detection. This likely reflects clinical guidelines which discourage microbiologic sampling of pharyngitis and impetigo [72]. This limits the review to reporting on clinical presentations likely representative of sGAS infection, without microbiological confirmation. This approach has clear clinical applications, though the meaningfulness of reported prevalences relies on past research showing sGAS as the major causative organism in impetigo and pharyngitis [10–12]. Diagnosis and reporting of sGAS infections was performed by different individuals across the studies, including doctors, nursing staff, community health officers, medical students and self-report. Few papers explored interobserver variability but it is plausible that the breadth of training and experience could contribute to variations in reported prevalence. Standardised nomenclature for classifying and recording impetigo and pharyngitis are needed to have greater certainty about sGAS burden. A standardised 'Skin Sore Severity Score' may be of benefit in future sGAS skin studies and it may be possible to optimise coding of encounters in primary care settings [47].

This paper has not included several papers included in past systematic reviews. With reference to the published protocol several articles were reviewed by the research team, given their inclusion in previously published evidence [19]. Discussion concluded that Shelby et al should be excluded from this analysis as the reported prevalence of impetigo was amongst children already diagnosed with trachoma, potentially confounding prevalence results if children with trachoma were more or less likely to have impetigo [73]. As previously discussed, data inclusive of isolated scabies infections were excluded from this burden analysis.

Ascertainment bias was certainly present as noted by the large number of impetigo studies within certain population groups (Aboriginal and Torres Strait Islander, remote) and a lack of the same population in the pharyngitis studies. It is difficult to conclude whether this increased attention and data collection represents a true increased impetigo or pharyngitis prevalence within these populations, or whether other factors such as prevalence of downstream disease (iGAS, RHD, ASPGN) and funding availability have led to this bias.

Other population factors limited our ability to present cohesive data. The non-standardisation of age groupings within the studies resulted in a need for statistical analysis to extrapolate age-related values, determining age-related prevalence that has obvious clinical applications. Furthermore, the geographical bias towards the Northern Territory / remote populations in

impetigo and Australia-wide or major city populations in pharyngitis limit our ability to compare the prevalence data across these populations. Similarly, many studies have focused on impetigo in remote living Aboriginal and Torres Strait Islander people, although relatively few have addressed the same epidemiology in Aboriginal and Torres Strait Islander people living in an urban centre, with a lack of any focused study on pharyngitis in Aboriginal and Torres Strait Islander people. Only one study, Health et al., directly compares Aboriginal and Torres Strait Islander and non-Indigenous population impetigo prevalence in the same urban school setting (22.2% vs 5.9%) [38]. There is a clear paucity of data for pharyngitis in remote regions and Aboriginal and Torres Strait Islander populations, as well as for impetigo in urban regions. This could be resolved with national notification of sGAS diseases. This geographical bias again limited the ability to compare populations within Australia.

Finally, an attempt was made to collect additional datapoints not presented in the results of this paper due to a lack of data in the included studies. Firstly, data on the social determinants of health (crowding, education, employment, income, nutrition, socioeconomic status and swimming) was rarely available and variable in collection method [27,30,47,48,55,57]. No single factor had enough statistical power to provide significant results. For each presentation we also attempted to collect data to further categorise disease severity and the presence of Streptococcal A bacteria. Again, the data was limited and varied in collection and definition, disallowing pooled analysis [35,43,50,53,56,59,60,62]. Future studies should consider comprehensive data collection methods including clear definition of disease, microbiological data in the presence of disease, presence of co-infection (particularly with scabies) and possible precipitating factors of disease inclusive of the social determinants of health, as recently published in the Strep A Vaccine Global Consortium (SAVAC) impetigo and pharyngitis guidelines [74].

This analysis indicates a substantial ongoing burden of sGAS in some settings in Australia. However, despite decades of research, substantial gaps in epidemiologic knowledge of sGAS remain. Specifically, there are knowledge gaps around pharyngitis prevalence, sGAS prevalence at different ages, in the eastern states and in urban settings. These issues cannot be resolved through further systematic reviews attempting to combine heterogenous studies over time. More small studies with bespoke definitions and methodologies are unlikely to add substantive epidemiologic value. In Aboriginal and Torres Strait Islander settings ad hoc studies risk contributing to the sense that communities are over-researched and underserved [75,76]. Rather, this analysis should help inform targeted research initiatives to address these knowledge gaps in conjunction with communities most affected by sGAS infections. Given there are populations globally with similarly high postGAS and iGAS burden [77,78], our recommendations can be generalised to focus on a cohesive large-scale and primary care based approach to define sGAS burden.

Gold standard epidemiologic data, guided by contemporaneous surveillance protocols such as the SAVAC impetigo and pharyngitis guideline [74], is likely to come from prospective sGAS surveillance studies in sentinel locations. In parallel, improved routine data collection through primary care may provide less detailed–but more sustainable–insights at a local, jurisdictional or national level. Improving the understanding of sGAS epidemiology is particularly urgent given progress towards Strep A vaccine development and the relevance of sGAS endpoints [79]. Equally urgently, cultural, research and clinical initiatives should address the apparent discrepancy between the community burden of sGAS disease and the prevalence reported within healthcare settings. Understanding the contributors to this gap and developing effective strategies to improve sGAS diagnosis, documentation and treatment is essential to reducing the burden of postGAS complications.

## Conclusion

In Australia, the burden of GAS spectrum diseases is significant, particularly for Aboriginal and Torres Strait Islander people. This paper presents a comprehensive review of the literature reporting on the epidemiology of two clinical presentations of sGAS disease (impetigo and pharyngitis) in Australia, representative of the major drivers of ARF and downstream postGAS disease. The included papers differed significantly in population and study design. Adherence of any future studies to standardised surveillance definitions may be of benefit.

Given the significant downstream burden of iGAS and postGAS disease resulting from preceding impetigo and/or pharyngitis in particular, an improved understanding of the epidemiology of this manifestation of GAS disease in Australia is critical for reducing disease prevalence across these varied endpoints. A prospective research agenda is needed, including strategies to standardise clinical definitions, nomenclature and knowledge gaps for specific populations. Implementation and evaluation of sGAS reduction approaches–such as environmental health, vaccination or enhanced primary care–depends on better quality longitudinal data. Communities, researchers and service providers are key stakeholders for achieving this vision.

## Supporting information

**S1 Fig. Consideration of bias: Funnel plots.**
(PDF)

**S1 Table. Summary of methods and baseline population characteristics of included articles.**
(PDF)

**S1 File. Protocol for the systematic review of the epidemiology of superficial Streptococcal A infections (skin and throat) in Australia.**
(PDF)

**S2 File. Preferred Reporting items for Systematic Reviews and Meta-Analyses (PRISMA) statement.**
(PDF)

**S3 File. Meta-analysis of Observational Studies in Epidemiology (MOOSE) guidelines.**
(PDF)

## Acknowledgments

This work was produced on Whadjuk Noongar Boodjar and Wongutha Country, and presents data collected from many Nations throughout Australia. We pay our respects to the people and Elders of these Nations, past, present and emerging.

This work was produced with the support of Dr Jeffrey Cannon, Dr Dylan Barth, The University of Western Australia Medical and Dental Librarians and The Telethon Kids Institute.

## Author Contributions

**Conceptualization:** Asha C. Bowen, Rosemary Wyber.

**Data curation:** Sophie Wiegele, Elizabeth McKinnon, Bede van Schaijik, Stephanie Enkel.

**Formal analysis:** Elizabeth McKinnon.

**Investigation:** Sophie Wiegele, Elizabeth McKinnon, Bede van Schaijik, Stephanie Enkel.

**Methodology:** Sophie Wiegele, Katharine Noonan, Asha C. Bowen, Rosemary Wyber.

**Project administration:** Sophie Wiegele.

**Supervision:** Katharine Noonan, Asha C. Bowen, Rosemary Wyber.

**Writing – original draft:** Sophie Wiegele, Elizabeth McKinnon, Rosemary Wyber.

**Writing – review & editing:** Sophie Wiegele, Elizabeth McKinnon, Bede van Schaijik, Stephanie Enkel, Katharine Noonan, Asha C. Bowen, Rosemary Wyber.

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
