## [Decision Letter · Decision Letter 0]

28 Feb 2023

PONE-D-22-28601The epidemiology of superficial Streptococcal A (impetigo and pharyngitis) infections in Australia: A systematic review.PLOS ONE

Dear Dr. Wiegele,

Thank you for submitting your manuscript to PLOS ONE. After careful consideration, we feel that it has merit but does not fully meet PLOS ONE’s publication criteria as it currently stands. Therefore, we invite you to submit a revised version of the manuscript that addresses the points raised during the review process.

We look forward to receiving your revised manuscript.

Kind regards,

Inge Roggen, M.D., Ph.D.

Academic Editor

PLOS ONE

Journal Requirements:

2. Please identify your study as "systematic review and meta-analysis" in the title of your manuscript.

“ncidental costs for this paper were supported by the NHMRC funded END RHD CRE (1080401). RW was supported an NHMRC Postgraduate Scholarship (1151165) and AB by an NHMRC Investigator Award (GNT1175509)”

4. Please update your submission to use the PLOS LaTeX template. The template and more information on our requirements for LaTeX submissions can be found at http://journals.plos.org/plosone/s/latex.

5**. **In your cover letter, please confirm that the research you have described in your manuscript, including participant recruitment, data collection, modification, or processing, has not started and will not start until after your paper has been accepted to the journal (assuming data need to be collected or participants recruited specifically for your study). In order to proceed with your submission, you must provide confirmation.

Reviewers' comments:

Reviewer's Responses to Questions

**Comments to the Author**

1. Does the manuscript adhere to the experimental procedures and analyses described in the Registered Report Protocol?

If the manuscript reports any deviations from the planned experimental procedures and analyses, those must be reasonable and adequately justified.

Reviewer #1: Yes

Reviewer #2: Yes

2. If the manuscript reports exploratory analyses or experimental procedures not outlined in the original Registered Report Protocol, are these reasonable, justified and methodologically sound?

A Registered Report may include valid exploratory analyses not previously outlined in the Registered Report Protocol, as long as they are described as such.

Reviewer #1: Yes

Reviewer #2: Yes

3. Are the conclusions supported by the data and do they address the research question presented in the Registered Report Protocol?

The manuscript must describe a technically sound piece of scientific research with data that supports the conclusions. The conclusions must be drawn appropriately based on the research question(s) outlined in the Registered Report Protocol and on the data presented.

Reviewer #1: Yes

Reviewer #2: Yes

4. Have the authors made all data underlying the findings in their manuscript fully available?

Reviewer #1: Yes

Reviewer #2: Yes

5. Is the manuscript presented in an intelligible fashion and written in standard English?

Reviewer #1: Yes

Reviewer #2: Yes

6. Review Comments to the Author

Please use the space provided to explain your answers to the questions above. (Please upload your review as an attachment if it exceeds 20,000 characters)

Reviewer #1: Work by Wiegele and colleagues appears methodologically sound and is well presented. If the paper does not bring real new information to the field, it provides an up-to-date review of superficial Strep A infection in Australia. The focus on (previously known) knowledge gaps with suggestions to improve future research is a noticeable strength.

Suggestions:

1) Consider adding a few more words about inclusion and exclusion criteria of the systematic review. Since the methods have been previously published (Wiegele, PLOS ONE, 2021) this reviewer found complicated to understand the scope without reading the methods paper.

2) The microbiological side of the inclusion/exclusion criteria remains confusing to this reviewer. In the methods paper it is stated that "Studies that were pathogen-specific, other than Streptococcal A, where participants numbered less than 20 and those describing a cohort not including children (defined as 0–16 years) will be excluded." In the discussion of the current manuscript, it is stated lines 396-399 that "Few studies reported on microbiological data from the patient cohort for either impetigo 50,53,56,59,60,62 (n=6) or pharyngitis 35,43,50 (n=3) disease resulting in insufficient power for statistical analysis based on GAS+ detection. This likely reflects clinical guidelines which discourage microbiologic sampling of pharyngitis and impetigo."

=> How do you therefore define "sGAS" without microbiological GAS confirmation? Please clarify in the methods, an apologies if I missed something.

This confusion (clinical skin infection without microbiological characterization versus GAS skin infection) seems to be present throughout the MS and especially in the discussion. Maybe define clearly each of the two situations with two different terms?

3) The discussion if logically focused on the Australian situation. However, please consider mentioning how the Australian experience can lead to improvement in the sGAS surveillance elsewhere in the world.

Minor Comments:

Line 93-95: "... not well described". Where? Do you mean in Australia? In at risk population? In the world? Please clarify.

Line 217 last word: "bar" should probably be replaced by "but"?

The conclusion is relatively lengthy with some repetition. Editing those would be great.

Reviewer #2: The article is interesting and well-written. This is a well-executed systematic review and meta-analysis.

It is striking that search strategies, title and abstract screen were completed by only one investigator (lines 132-133). Authors should justify why the title and the abstract screen were not done by at least two investigators independently and to what extent this might affect the inclusion of the articles.

7. PLOS authors have the option to publish the peer review history of their article (what does this mean?). If published, this will include your full peer review and any attached files.

Reviewer #1: No

Reviewer #2: No

---

## [Author Response · Author response to Decision Letter 0]

29 May 2023

Thank you for your feedback and suggestions on this work. The points offered for change will improve and clarify the research we are presenting here. We will break down our response to be comment specific.

Reviewer 1

1) We agree that it would be important to further clarify inclusion and exclusion criteria early. An extended inclusion and exclusion criteria has been added to lines 116- 122.

2) The aim of the systematic review is to document the epidemiology of the clinical presentations of probable sGAS infections in Australia. Our current practice is to treat these infections based on clinical presentation without taking microbiological samples or relying on these results to guide treatment. Therefore, by adhering to best practice, there will be no microbiological confirmation of sGAS infection and a systematic review focused on microbiologically confirmed sGAS infection would likely under-represent the overall burden of disease. A systematic review reporting prevalence of clinical presentations has greater clinical application and meaningful contribution to the current literature.

We agree that this needs to be clarified in the paper. Current explanations have been made line 91-2, 409-12. We have expanded on this point in line 93-95 and reflected on this limitation in lines 408-11.

3) Thank you for this feedback. We agree that placing our systematic review in a global context may be useful to clinicians and researchers. 

We have addressed this 471-3 in the discussion.

Minor comments

- Point regarding current throat sGAS research has been clarified in line 95

- Line 229 – use of ‘bar’ here meaning ‘except for’. Happy to change to ‘but’ for clarity.

Reviewer 2

1) The screening process has been clarified in line 141-2.

---

## [Editor Report · Decision Letter 1]

19 Jun 2023

The epidemiology of superficial Streptococcal A (impetigo and pharyngitis) infections in Australia: A systematic review.

PONE-D-22-28601R1

Dear Dr. Wiegele,

We’re pleased to inform you that your manuscript has been judged scientifically suitable for publication and will be formally accepted for publication once it meets all outstanding technical requirements.

Kind regards,

Inge Roggen, M.D., Ph.D.

Academic Editor

PLOS ONE
---

## [Editor Report · Acceptance letter]

25 Jul 2023

PONE-D-22-28601R1 

The epidemiology of superficial Streptococcal A (impetigo and pharyngitis) infections in Australia: A systematic review. 

Dear Dr. Wiegele:

I'm pleased to inform you that your manuscript has been deemed suitable for publication in PLOS ONE. Congratulations! Your manuscript is now with our production department. 

Kind regards, 

on behalf of

Prof. Inge Roggen 

Academic Editor

PLOS ONE